# Comparison of Conventional versus Modified Preperitoneal Pelvic Packing in Patients with Bleeding Pelvic Fractures: A Single-Center Retrospective Pilot Study

**DOI:** 10.3390/jcm13144062

**Published:** 2024-07-11

**Authors:** Sebeom Jeon, Byungchul Yu, Gil Jae Lee, Min A Lee, Jungnam Lee, Kang Kook Choi

**Affiliations:** 1Department of Trauma Surgery, Gachon University Gil Medical Center, Incheon 21565, Republic of Korea; dsjeonse@gmail.com; 2Department of Traumatology, College of Medicine, Gachon University, Incheon 21565, Republic of Korea; kane2123@gilhospital.com (B.Y.); nonajugi@gilhospital.com (G.J.L.); jwh@gilhospital.com (M.A.L.); jnlee@gilhospital.com (J.L.)

**Keywords:** pelvic bones, hemorrhage, mortality, surgical procedures, embolization

## Abstract

**Background**: Bleeding pelvic fractures have high mortality rates, primarily due to severe hemorrhage. Treatment options include mechanical stabilization based on preperitoneal pelvic packing (PPP), resuscitative endovascular balloon occlusion of the aorta, and angioembolization (AE). The bilateral preperitoneal approach, which uses three pads on each side, is the conventional PPP method. We aimed to compare the bilateral preperitoneal approach with a modified approach, involving selectively packing only heavily bleeding areas, in terms of clinical outcomes and mortality risks. **Methods:** We included patients who underwent PPP and compared the outcomes between conventional (three sponges placed on each side) and modified PPP (selective packing of critical areas). The primary outcome was 30-day mortality; the secondary outcomes included 24 h mortality, pelvic complications, and transfusion requirements. Univariate and multivariate analyses were performed to determine risk factors for 30-day and 24 h mortality. **Results:** Among the 47 included patients, 19 and 28 underwent conventional and modified PPP, respectively. There were no significant between-group differences in the 24 h (26.3% vs. 42.9%, *p* = 0.247) and 30-day mortality rates (47.4% vs. 60.7%, *p* = 0.366). Using univariate and multivariate analyses, initial lactate levels and the decision to perform AE were found to be significant risk factors for mortality. However, the selected PPP method was not a risk factor for 30-day mortality (odds ratio [OR], 2.22; 95% confidence interval [CI], 0.27–18.26; *p* = 0.457) or 24 hr mortality (OR, 1.77; 95% CI, 0.24–13.19; *p* = 0.557). **Conclusions:** The modified PPP method may be considered in patients with bleeding pelvic fractures for effective bleeding control while minimizing potential complications associated with the conventional PPP.

## 1. Introduction

The mortality rate among patients with hemodynamically unstable pelvic fractures ranges between 28% and 42% [1,2,3]. These fractures are often associated with severe hemorrhage and retroperitoneal bleeding, which may lead to fatal outcomes [4]. Early hemorrhage control is crucial in patients with pelvic fractures complicated by bleeding [5]. Pelvic angioembolization (AE) is commonly used to manage arterial bleeding associated with pelvic fractures [6]; resuscitative endovascular balloon occlusion of the aorta (REBOA) may be used as an adjunctive method [7,8]. The mechanical stabilization of pelvic bleeding is performed using external fixation and pelvic binders [9,10]. Further, preperitoneal pelvic packing (PPP) is essential for controlling venous bleeding in the pelvic cavity and achieving a tamponade effect within the preperitoneal space [11]. Although these treatments can be applied alone or in combination, there remains no standardized approach [12]. In patients with hemodynamically unstable pelvic fractures, the main cause of bleeding in 80–90% of cases is venous hemorrhage [13,14]. This suggests that PPP could be a valuable procedure for initial bleeding control in patients with bleeding pelvic fractures [15].

PPP was initially introduced as a method involving the placement of three pads on each side of the pelvic rim, totaling six pads [16,17]. However, the dissection of the retroperitoneal space on the non-bleeding side for PPP may increase the risk of complications, including bleeding and infection [18,19]. Accordingly, we developed a modified PPP method that could selectively compress the bleeding site. The outcomes of selectively packing only the bleeding side in PPP remain unclear.

We hypothesized that a modified PPP approach targeting only the bleeding sites could achieve hemostasis that is equivalent to or more effective than the traditional bilateral method, with fewer complications such as surgical site infections. This study aimed to compare the clinical outcomes between these two PPP methodologies and explore whether the differences in these approaches could contribute to risk factors for mortality.

## 2. Materials and Methods

### 2.1. Patient Population and Data Collection

This retrospective cohort study was conducted at a single center using data from a regional trauma registry. We included patients admitted to the Regional Trauma Center at Gachon University Gil Hospital from January 2014 to December 2022. Specifically, we included patients who underwent PPP to treat hemorrhaging pelvic fractures. We excluded patients who sustained injuries through penetrating mechanisms, those who received only non-operative management, or those who only underwent AE or REBOA. Additionally, we excluded patients who were declared dead upon arrival at the emergency department. Before study initiation, an ethical review was conducted by the Institutional Review Board or Ethics Committee, which subsequently granted formal approval (GDIRB2024-152).

### 2.2. Preperitoneal Pelvic Packing Techniques

The Denver group has consistently advocated for the preperitoneal approach as the preferred technique for bilateral PPP (Figure 1) [17].

This approach is currently widely accepted in the medical community. According to their proposed method, three standard surgical laparotomy pads are deeply inserted beneath the bladder on each side, at the edges of the pelvis, which apply pressure and control hemorrhage. Subsequently, this sequence is repeated on the opposite side. In our study, the patients who underwent PPP using this technique were categorized into the conventional PPP group. Conversely, we defined the modified PPP group as patients who received treatment with a focus only on the main bleeding area. In the modified PPP group, the patients were treated by placing sponges unilaterally, placing sponges bilaterally but with a greater quantity on one side due to severe bleeding, or placing sponges solely on the anterior side of the pelvic brim (Figure 2).

Patients who had unilateral pelvic fractures on initial emergency room pelvic radiography, unilateral contrast extravasation on abdominal computed tomography, or unilateral bleeding during surgery following a midline incision were included in the modified PPP group. Surgical records and imaging were meticulously reviewed to ensure that this technique was applied to all relevant patients.

### 2.3. Outcomes

We collected the following data from electronic medical records: patient demographics; injury mechanisms; injury severity scores; associated injuries; hemodynamic status; initial Glasgow Coma Scale (GCS) score; baseline lactate levels; pelvic fracture type (Young–Burgess classification); use of interventional radiography and AE; use of REBOA; and mechanical pelvic stabilization.

The primary outcomes were the 30-day and 24 h mortality rates. The secondary outcomes included hospital length of stay (LOS), intensive care unit LOS, complications, volume of packed red blood cells (RBCs), and fresh frozen plasma administered within 4 h and 24 h. Complications were categorized as pelvic or non-pelvic. Pelvic complications included surgical site infections and the need for repacking, while non-pelvic complications included pneumonia, central line-associated bloodstream infections (CLABSIs), venous thromboembolism (VTE), and urinary tract infections. In cases of reoperation for removing surgical pads inserted during the initial PPP, wherein bleeding was determined not to be sufficiently controlled, repacking was performed at the surgeon’s discretion.

### 2.4. Statistical Analysis

Categorical variables were analyzed using the chi-square test and Fisher’s exact test. Continuous variables were analyzed using the Student’s *t*-test and Mann–Whitney U test. Due to the small group sizes, normality tests were conducted for all variables. Regarding variables that did not follow a normal distribution, non-parametric tests were applied. Categorical variables are presented as frequencies and percentages. Continuous variables are described using the median and either the standard deviation (SD) or the interquartile range, based on their distribution. We dichotomized continuous variables at clinically relevant thresholds (age > 55 years, systolic blood pressure < 90 mmHg, heart rate > 120 bpm, GCS score < 9, Injury Severity Score (ISS) > 25, Abbreviated Injury Scale (AIS) score ≥ 3, and red blood cell transfusion volumes > 10 units). Given the small group sizes, non-parametric tests, including Fisher’s exact test and the Mann–Whitney U test, were applied to non-normally distributed variables, as confirmed by tests for normality. Statistical significance was set at a *p*-value less than 0.05. Univariate and multivariate analyses were employed to explore risk factors for 30-day and 24 h mortality. Statistical analyses were performed using SPSS for Windows, version 22.0 (SPSS Inc.; Chicago, IL, USA).

## 3. Results

During the study period, we identified 47 patients with bleeding pelvic fractures who were eligible for the PPP protocol. Among them, 19 and 28 patients were treated with the conventional and modified PPP approach, respectively.

Age, sex, initial vital signs in the emergency room, the ISS, and the pelvic fracture type showed no significant between-group differences. However, there were significant between-group differences in the mechanism of injury (*p* = 0.01). The most common mechanisms of injury in the conventional and modified PPP groups were pedestrian accidents (conventional PPP vs. modified PPP = 42.1% vs. 7.1%) and falls (conventional PPP vs. modified PPP = 21.1% vs. 67.9%), respectively. The GCS scores were significantly higher in the conventional PPP group (median 12, range 9–15 vs. median 5, range 3–13; *p* = 0.008) than in the modified PPP group. Furthermore, the proportion of patients with a head AIS score ≥ 3 was significantly lower in the conventional PPP group than in the modified PPP group (15.8% vs. 46.4%, *p* = 0.03) (Table 1).

There was no significant between-group difference in the 24 h (26.3% vs. 42.9%, *p* = 0.247) and 30-day mortality rates (47.4% vs. 60.7%, *p* = 0.366). Surgical site infections (SSIs) occurred in four and three patients in the conventional and modified PPP groups, respectively (21.2% vs. 10.7%, *p* = 0.417). Moreover, PPP re-implementation was necessary in two and five patients in the conventional and modified PPP groups, respectively (10.5% vs. 17.9%, *p* = 0.685). There were no significant between-group differences in pelvic complications. The rate of extra-pelvic complications, including infection-related pneumonia, CLABSIs, and urinary tract infections, was higher in the conventional PPP group. However, there were no significant between-group differences. Both groups had one case each of VTE. There was no significant between-group difference in the amount of transfusion required within the first 4 h. However, the conventional PPP group required significantly more transfusions within 24 h (median 8, range 3–21 vs. median 5, range 1–12, *p* = 0.011) (Table 2).

The univariate analysis revealed that the initial lactate level (odds ratio [OR], 1.18; 95% confidence interval [CI], 1.01–1.39; *p* = 0.037), a GCS score ≤ 9 (OR, 4.87; 95% CI, 1.28–18.57; *p* = 0.020), and the presence of a pelvic AE (OR, 1.01; 95% CI, 0.97–16.77; *p* = 0.054) were significant predictors of 24 h mortality. The multivariate analysis revealed that a GCS score ≤ 9 (OR, 3.48; 95% CI, 0.84–14.31; *p* = 0.083) and the presence of pelvic AE (OR, 1.01; 95% CI, 0.97–16.77; *p* = 0.054) remained significant predictors. However, the choice of PPP method was not a significant risk factor (OR, 2.14; 95% CI, 0.28–16.43; *p* = 0.464) (Table 3).

The univariate analysis revealed the following risk factors for 30-day mortality: the initial lactate levels (OR, 1.17; 95% CI, 1.01–1.37; *p* = 0.042), a GCS score ≤ 9 (OR, 5.54; 95% CI, 1.56–19.61; *p* = 0.008), and the presence of pelvic arterial embolization (OR, 7.20; 95% CI, 1.38–37.35; *p* = 0.019). The multivariate analysis with adjustment for confounding variables confirmed that a GCS score ≤ 9 remained a significant risk factor (OR, 4.07; 95% CI, 1.07–15.39; *p* = 0.038). Moreover, the choice of PPP method was not significantly associated with 30-day mortality (OR, 1.96; 95% CI, 0.34–11.15; *p* = 0.447) (Table 4).

## 4. Conclusions

This study investigated and compared clinical outcomes associated with conventional and modified PPP techniques in the management of hemodynamically unstable pelvic fractures. To the best of our knowledge, this is the first study to explore the clinical outcomes of the modified PPP technique.

Logothetopulos described a method for managing extensive pelvic bleeding in 1926 [20], later modified by Foreman in 1995 to a preperitoneal approach [21]. The Denver group introduced ‘direct’ PPP for retroperitoneum access via a suprapubic midline incision [16,17]. Observational studies indicated that this method could improve survival in patients with hemodynamically unstable pelvic fractures [22,23,24], though they used bilateral packing without considering bleeding patterns. Bleeding in such fractures often varies with fracture pattern and external force impact, typically being unilateral. We hypothesized that dissecting non-bleeding tissue for packing could induce unnecessary bleeding and infection, supported by studies on surgeries with extraperitoneal access [18,25]. The multivariate analysis revealed that the PPP technique itself was not a significant risk factor for mortality. This underscores the concept that patient-specific factors and the severity of injury play a more crucial role in determining outcomes than the choice of surgical technique. This suggests that the modified PPP technique can achieve effectiveness comparable to that of the conventional method in terms of mortality. The fact that the modified PPP technique does not significantly increase mortality rates despite being a selective approach allows surgeons to tailor treatment strategies more flexibly based on individual patient needs, particularly in cases where the bleeding source is clearly localized. In our results, although not statistically significant, the 24 h and 30-day mortality rates were higher in the modified PPP group. However, the modified PPP group had significantly higher lactate levels and significantly lower GCS scores, suggesting that this group included more patients with higher physiological and neurological severity. Despite this, the mortality rates between the two groups did not differ significantly. The higher mortality rate in this study was likely due to urgent PPP in patients with severe injuries and unstable vital signs. At our center, AE is prioritized if the patient is stable and it is possible within an hour. This may explain the higher mortality rate compared with that in centers prioritizing PPP. With 87% of patients having an ISS ≥ 25, our mortality rate is consistent with the expected outcomes in patients with high ISSs [26].

Several retrospective observational studies have shown that the implementation of standardized multidisciplinary clinical guidelines, including early surgical management with pelvic external fixation and direct PPP, in hypotensive patients with hemodynamically and mechanically unstable pelvic ring injuries significantly reduced transfused blood products and post-injury mortality [27,28]. In our study, fewer RBCs were transfused within 24 h in the modified PPP group. Although not generalizable, the selective packing of only suspected bleeding areas may at least prevent the need for more transfusions. The modified PPP approach did not require increased transfusions, which suggests that it may avoid unnecessary bleeding. One of the most devastating complications of the treatment in patients with a hemodynamically unstable pelvic floor is the development of VTE, which includes deep vein thrombosis and pulmonary embolism. PPP has been shown to increase the incidence of VTE [29,30]. PPP may cause VTE through a decrease in blood flow velocity due to shock, a delay in thromboprophylaxis, and an increase in blood transfusion volume [29]. Given our hypothesis that selective pelvic packing can prevent unnecessary bleeding and reduce transfusion volume within 24 h, selective pelvic packing may possibly reduce the incidence of VTE by reducing the transfusion volume and allowing for prompt antithrombotic therapy.

Another significant complication of PPP is SSI [31]. The occurrence of SSI following PPP interventions has been reported to range between 12% and 34% [22,32]. In our cohort, the overall incidence of SSI was 15%. Although not statistically significant, the conventional treatment group exhibited a higher rate of SSI. There has been limited research on the risk factors for post-PPP SSI. However, some studies have suggested an increased rate of SSI when repacking is necessary. In this study, there was no significant between-group difference in the repacking rates, indicating that the modified PPP method did not increase the incidence of SSI. Additionally, abdominal compartment syndrome (ACS) is one of the major complications that can occur after PPP; packing more gauze could result in increased intra-abdominal pressure. Therefore, the conventional PPP technique, which involves more gauze packing, has the potential to cause ACS more frequently compared with the modified PPP technique. However, this study did not analyze the incidence of ACS; further research is needed on this aspect [33,34].

This study has several limitations. First, since this was a single-center retrospective study based on data from one regional trauma center, its generalizability may be limited due to varying practices across institutions. Second, this study was conducted with a relatively small sample size. As a pilot study, one of the achievements of this research is that it demonstrated the safety and effectiveness of the modified PPP method, providing a basis for conducting large-scale studies in the future. Third, we did not investigate the types of pelvic fractures. However, our study focused on patients with hemorrhagic pelvic fractures rather than other types of pelvic fractures; all patients with hemorrhagic pelvic fractures were included throughout the study period. Fourth, the modified PPP technique was introduced by our institution due to concerns about the additional damage caused by dissecting through ‘normal tissue’ using the conventional PPP technique. Therefore, during the initial use of this method, there may have been issues with standardization regarding the choice between the conventional PPP and modified PPP techniques. Finally, although no significant differences were found in the physiological and injury severity between the two groups, unknown or unmeasured confounding variables may have been present.

In conclusion, while both the conventional and modified PPP techniques are effective in managing severe pelvic bleeding, the modified approach may allow for more precise, patient-specific treatment without compromising safety or efficacy. Future studies are warranted to confirm these findings and potentially revise clinical guidelines for accommodating the flexibility offered by the modified PPP method, particularly in centers where individualized patient care is paramount.

## Figures and Tables

**Figure 1 jcm-13-04062-f001:**
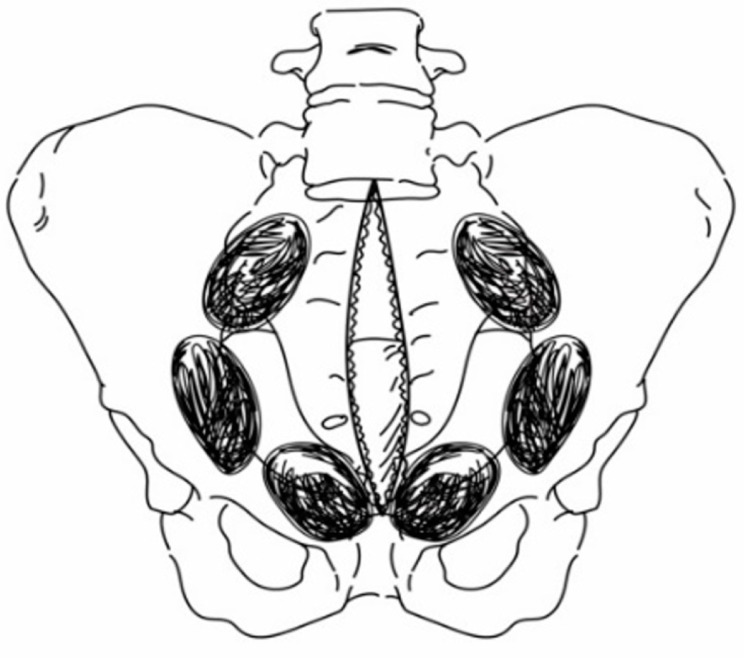
The conventional PPP technique involves the placement of three standard laparotomy pads deep with the preperitoneal space on both sides of the bladder. PPP: preperitoneal pelvic packing.

**Figure 2 jcm-13-04062-f002:**
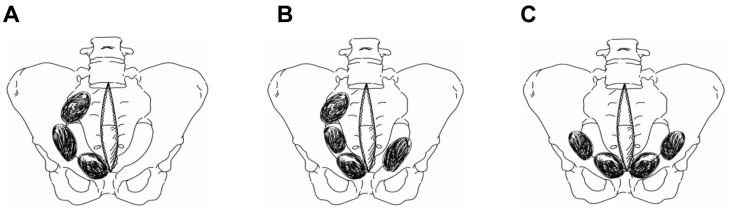
In the modified PPP approach, the laparotomy pads are strategically positioned either solely on one side (**A**), with an increased number of pads on one side (**B**), or exclusively along the anterior aspect of the pelvic brim (**C**). PPP: preperitoneal pelvic packing.

**Table 1 jcm-13-04062-t001:** Clinical characteristics of patients in the two groups.

Characteristics	Conventional PPP (n = 19)	Modified PPP (n = 28)	*p* Value
Age, years, mean ± SD	49.5 ± 21.6	54.3 ± 16.3	0.412
Gender			0.589
Male, *n* (%)	13 (68.4)	17 (60.7)	
Mechanism, *n* (%)			0.01
MVC	1 (5.3)	1 (3.6)	
MCC	1 (5.3)	0 (0.0)	
AVP	8 (42.1)	2 (7.1)	
Fall	4 (21.1)	19 (67.9)	
Others	5 (26.3)	6 (21.4)	
Vital signs at ED			
SBP, mmHg, mean ± SD	79.5 ± 24.6	77.7 ± 37.1	0.854
HR, bpm median [IQR]	98.0 [90.0–140.0]	119.0 [103.0–125.0]	0.704
GCS, median [IQR]	12 [9–15]	5 [3–12]	0.008
Lactate, mmol/L, mean ± SD	6.6 ± 3.6	8.7 ± 4.4	0.084
AIS ≥ 3, *n* (%)			
Head ≥ 3	3 (15.8)	13 (46.4)	0.030
Chest ≥ 3	12 (63.2)	19 (67.9)	0.739
Abdomen ≥ 3	14 (73.7)	15 (53.6)	0.164
Extremity ≥ 3	19 (100.0)	22 (78.6)	0.031
ISS, mean ± SD	39.4 ± 11.7	38.6 ± 12.7	0.828
ISS > 25, *n* (%)	18 (94.7%)	23 (82.1%)	0.378
Pelvic fracture type (Young–Burgess classification), *n* (%)			0.463
APC type II	1 (5.3)	2 (7.1)	
APC type III	4 (21.1)	4 (14.3)	
LC type I	1 (5.3)	6 (21.4)	
LC type II	7 (36.8)	5 (17.9)	
LC type III	3 (15.8)	7 (25.0)	
VS type	1 (5.3)	3 (10.7)	
Combined	2 (10.5)	1 (3.6)	
Pelvic AE, *n* (%)	11 (57.9%)	22 (78.6%)	0.128
REBOA, *n* (%)	2 (10.5%)	8 (28.6%)	0.138

MVC, Motor Vehicle Crash; MCC, Motorcycle Crash; AVP, Automobile Versus Pedestrian; ED, emergency department; SBP, systolic blood pressure; HR, heart rate; GCS, Glasgow Coma Scale; AIS, Abbreviated Injury Scale; ISS, Injury Severity Score; APC, Anterior–Posterior Compression; LC, Lateral Compression; VS, Vertical Shearing; AE, angioembolization; REBOA, resuscitative endovascular balloon occlusion of the aorta.

**Table 2 jcm-13-04062-t002:** Clinical outcomes and complications according to the type of PPP.

	Conventional PPP (n = 19)	Modified PPP (n = 28)	*p* Value
Mortality, *n* (%)			
24 h mortality	5 (26.3%)	12 (42.9%)	0.247
30-day mortality	9 (47.4%)	17 (60.7%)	0.366
Length of stay, days, median [IQR]			
ICU LOS	8 [2–17]	4 [1–19]	0.262
Hospital LOS	26 [2–61]	4 [1–42]	0.095
Pelvic complications, *n* (%)	6 (31.6)	8 (28.6)	0.975
SSI	4 (21.2%)	3 (10.7%)	0.417
Repacking	2 (10.5%)	5 (17.9%)	0.685
Non-pelvic complications, *n* (%)	12 (63.1)	8 (28.6)	0.061
Pneumonia	7 (36.8%)	5 (17.9%)	0.143
CLABSIs	4 (21.2%)	2 (7.1%)	0.204
VTE	1 (5.3%)	1 (3.6%)	1.000
Transfusion			
pRBCs, units/4 h, mean ± SD	11.2 ± 9.6	12.6 ± 7.2	0.563
Plasma, units/4 h, mean ± SD	4.6 ± 4.0	5.1 ± 4.6	0.737
pRBCs, units/24 h, median [IQR]	9 [3–26]	4 [0–10]	0.011
Plasma, units/24 h, median [IQR]	8 [3–21]	5 [1–12]	0.187

ICU, intensive care unit; LOS, length of stay; SSI, surgical site infection; CLABSIs, central line-associated blood stream infections; SD, standard deviation; VTE, venous thromboembolism; PPP, preperitoneal pelvic packing; pRBCs, packed red blood cells.

**Table 3 jcm-13-04062-t003:** Univariate and multivariate analyses for risk factors of 24 h mortality.

Factors	Univariate (24-Day Mortality)			Multivariate (24-Day Mortality)		
	OR	95% CI	*p* Value	OR	95% CI	*p* Value
Age > 55	1.60	0.47–5.47	0.450			
Sex (male/female)	1.40	0.41–4.78	0.591			
SBP < 90 mmHg	2.33	0.54–10.05	0.255			
HR > 120 bpm	1.68	0.51–5.60	0.393			
Lactate	1.18	1.01–1.38	0.037	1.13	0.94–1.34	0.191
GCS ≤ 9	4.87	1.28–18.57	0.020	2.06	0.42–10.09	0.373
ISS > 25	1.70	0.38–7.50	0.486			
Pelvic AE (Y/N)	0.18	0.05–0.69	0.012	0.10	0.97–16.77	0.054
REBOA (Y/N)	2.08	0.50–8.60	0.310			
Pelvic complications (Y/N)	6.85	0.78–59.81	0.081			
pRBCs > 10	1.83	0.53–6.24	0.332			
Modified PPP	2.10	0.59–7.44	0.251	4.81	0.89–26.04	0.068

SBP, systolic blood pressure; HR, heart rate; GCS, Glasgow Coma Scale; ISS, Injury Severity Score; AE, angioembolization; REBOA, resuscitative endovascular balloon occlusion of the aorta; pRBCs, packed red blood cells; PPP, preperitoneal pelvic packing.

**Table 4 jcm-13-04062-t004:** Univariate and multivariate analyses of risk factors for 30-day mortality.

Factors	Univariate (30-Day Mortality)			Multivariate (30-Day Mortality)		
	OR	95% CI	*p* Value	OR	95% CI	*p* Value
Age > 55	1.70	0.53–5.50	0.376			
Sex (male/female)	2.40	0.68–8.50	0.175			
SBP < 90 mmHg	1.88	0.52–6.85	0.336			
HR > 120 bpm	1.39	0.43–4.49	0.579			
Lactate	1.17	1.01–1.37	0.042	1.14	0.96–1.36	0.138
GCS < 9	5.54	1.56–19.61	0.008	4.07	1.07–15.39	0.038
ISS > 25	1.17	0.30–4.54	0.824			
Pelvic AE (Y/N)	0.02	0.03–0.72	0.019	0.20	0.04–1.12	0.067
REBOA (Y/N)	1.98	1.44–8.89	0.371			
Pelvic complications (Y/N)	2.46	0.59–10.28	0.216			
pRBCs > 10	2.08	0.64–6.74	0.258			
Modified PPP	2.00	0.61–6.56	0.252	2.07	0.38–11.36	0.404

SBP, systolic blood pressure; HR, heart rate; GCS, Glasgow Coma Scale; ISS, Injury Severity Score; AE, angioembolization; REBOA, resuscitative endovascular balloon occlusion of the aorta; pRBCs, packed red blood cells; PPP, preperitoneal pelvic packing.

## Data Availability

The data presented in this study are available upon request from the corresponding author. The data are not publicly available due to privacy or ethical restrictions.

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
