# Peer review of "Comparison of Conventional versus Modified Preperitoneal Pelvic Packing in Patients with Bleeding Pelvic Fractures: A Single-Center Retrospective Pilot Study"

_jcm, 2024, doi:10.3390/jcm13144062_

Round 1

Reviewer 1 Report

Comments and Suggestions for Authors

I thank the authors for the opportunity they have given me to read this interesting paper and I congratulate them for the work they have done.
I have important doubts from a methodological point of view.
The authors have collected patients between 2014-2022. Have all patients who met inclusion criteria been included?
How have you calculated the sample size necessary to demonstrate significant differences in the main variable? The authors present results from 47 patients (19-28) with a number clearly not balanced between groups. Could this not influence the results? What was the calculation of the number of patients needed in each group?
This brings me to the main result (30-day mortality) there is a notable difference between groups 47.4% and 60.7% (more than 13% mortality in modified PPP), although the result is not statistically significant, right? Do the authors consider it to be clinically relevant?
It could be that statistical significance was not reached because the sample size necessary to reveal these differences was not reached.
Without a prior sample size calculation the data may be biased.
The appreciation of one technique with a mortality greater than 13% at 30 days and greater than 16% at 24 hours over another, I do not believe demonstrates their equal safety. This aspect should be discussed in depth, especially if it has not been demonstrated to have a sufficient sample size (overall and by groups).
Other methodological aspects that concern me are the inclusion criteria, they should be well defined (as are the exclusion criteria) and, more importantly, what criteria were standardized to choose PPP or mPPP. What did the surgeon base on choosing one technique or another? This aspect should be clearly defined.
The authors dichotomize some variables. What was the basis for choosing these cut-off points and not others? It should be well described.
Continuous variables are presented as means and standard deviations. Did the authors check the normality of the variables?
It is strange that they all present normality due to such a small sample size. Otherwise, they should use medians and interquartile ranges and non-parametric hypothesis tests.
Authors must add the meaning of the abbreviations in each table (e.g. mvc, mcc, avp, etc.)

Author Response

Comment1 [The authors have collected patients between 2014-2022. Have all patients who met inclusion criteria been included?]

Response1 [We thank you for this query. We included all patients who met the inclusion criteria we proposed.]

Comment2 [How have you calculated the sample size necessary to demonstrate significant differences in the main variable? The authors present results from 47 patients (19-28) with a number clearly not balanced between groups. Could this not influence the results? What was the calculation of the number of patients needed in each group?]

Response2 [We agree with you that determining the sample size is crucial for research. However, the appropriate sample size was typically determined by prior research. In this study, there were no relevant prior studies to refer to. This study serves as a pilot study; we suggest that more extensive follow-up research will be necessary in the future. We have added this point to the Discussion section.]

Comment3 [This brings me to the main result (30-day mortality) there is a notable difference between groups 47.4% and 60.7% (more than 13% mortality in modified PPP), although the result is not statistically significant, right? Do the authors consider it to be clinically relevant?

It could be that statistical significance was not reached because the sample size necessary to reveal these differences was not reached.

Without a prior sample size calculation the data may be biased.

The appreciation of one technique with a mortality greater than 13% at 30 days and greater than 16% at 24 hours over another, I do not believe demonstrates their equal safety. This aspect should be discussed in depth, especially if it has not been demonstrated to have a sufficient sample size (overall and by groups).]

Response3 [Thank you for pointing out this important issue. There can be several interpretations of these results. First, it might be due to coincidence caused by the small sample size. However, the modified PPP group had significantly higher lactate levels and significantly lower GCS scores. This indicated that the modified PPP group included patients who were physiologically and neurologically worsening conditions. Lactate levels and GCS are important prognostic factors in patients with trauma. Additionally, the modified PPP group had a higher frequency of TAE or REBOA procedures (although not statistically significant), which also suggests that more severely ill patients might have been included in the modified PPP group. Therefore, we are cautious in interpreting the higher mortality rate of modified PPP compared with the conventional PPP as an indication that the modified PPP is an inferior procedure. We have added this point to lines 230 to 234 of the Discussion section of the manuscript.]

Comment4 [Other methodological aspects that concern me are the inclusion criteria, they should be well defined (as are the exclusion criteria) and, more importantly, what criteria were standardized to choose PPP or modified PPP. What did the surgeon base on choosing one technique or another? This aspect should be clearly defined.]

Response4 [The modified PPP was introduced by our institution due to concerns about the additional damage caused by dissecting through 'normal tissue' with the conventional PPP. Therefore, during the initial use of this method, there may have been issues with standardization regarding the choice between the conventional PPP and modified PPP. We consider this to be one of the limitations of our study and have added it to lines 278 to 282 of the limitations section. As mentioned in the main text, if bleeding is localized to one side of the pelvic cavity, we perform packing only on the affected side. However, throughout the study period, it is possible that the conventional PPP was used more frequently during the early stages.]

Comment5 [The authors dichotomize some variables. What was the basis for choosing these cut-off points and not others? It should be well described.]

Response5 [We thank you for these comments that were very accurate and helpful.

  1. SBP < 90mmHg is a criterion for hemorrhagic shock. This is a widely recognized threshold that has been commonly used in previous trauma studies to identify hemorrhagic shock. We proposed this criterion to analyze whether the presence of hemorrhagic shock in the initial emergency department phase was a risk factor for mortality. Similarly, HR > 120 bpm was also considered.
  2. A GCS score of 9 or below indicates a life-threatening critical condition requiring prompt and aggressive medical intervention. This score serves as a crucial parameter for the initial assessment, ongoing monitoring, and prognosis in patients with trauma.
  3. An ISS of 25 or above indicates a patient with severe multiple injuries. Although an ISS of 16 is sometimes used as a threshold, we set 25 as the threshold for our study population because they are patients with hemorrhagic pelvic fractures, which are of higher severity. This score indicates a high risk of mortality and morbidity, necessitating immediate and intensive multidisciplinary treatment. It is used as an essential tool for trauma assessment, clinical decision-making, resource allocation, and long-term care planning. Comparisons with previous studies and references to these criteria are mentioned in the discussion.
  4. The criterion of more than 10 units of red blood cell transfusion is based on the definition of massive transfusion, which is defined as transfusing more than 10 units of red blood cells within 24 h.]

Comment6 [Continuous variables are presented as means and standard deviations. Did the authors check the normality of the variables?

It is strange that they all present normality due to such a small sample size. Otherwise, they should use medians and interquartile ranges and non-parametric hypothesis tests.]

Response6 [We recognized from the beginning of the study that there was a limitation due to the small sample size. Therefore, we thoroughly performed normality tests for testing all variables. If a variable did not follow normality, we conducted the Mann-Whitney U test. However, we initially presented the results as mean and standard deviation. We agree with you regarding the comments and have revised the presentation of non-normally distributed variables to median and interquartile range based on the normality tests. And I have added the statistical methods based on the normality of the variables to lines 118 to 122 of the manuscript. Thank you for your detailed feedback.]

Comment7 [Authors must add the meaning of the abbreviations in each table (e.g. mvc, mcc, avp, etc.)]

Response7 [Thank you for your important comments. We have revised the text according to your suggestions.]

Reviewer 2 Report

Comments and Suggestions for Authors

Thank you for your interesting manuscript regarding the comparison of the two different techniques for preperitoneal pelvic packing in patients with pelvic fractures. It was interesting, but I have made some suggestions.

Abstract: short and on-point

Introduction:

Line 35: Please adjust for the reference list. There are three references, but the range consists of only two numbers, which may be considered an incorrect citation style.

General: Short and on-point. Quick and well-written introduction to the topic.

Materials and Methods:

Were bilateral fractures excluded?

Please describe the types of pelvic fractures included. Which classification system was used?

General: Except for the aforementioned aspects, this is a well-written section.

Results:

Line 144: What were the reasons for reimplementation?

Discussion:

Line: 168-189: this section is redundant and wordy. PPP is primarily performed in unstable patients. Please shorten the whole paragraph. (Line 186-194)

Line: 195-215: this section is more suitable for the introduction section and not the discussion section. Please rewrite and shorten it significantly and discuss the findings mentioned in the manuscript.

Thank you for allowing me to review this manuscript on an infrequent topic. It is well written and in proper English, but has some major flaws. Therefore  I would recommend this submission after meticulous 

Sincerely

Author Response

Introduction:

Comment1 [Line 35: Please adjust for the reference list. There are three references, but the range consists of only two numbers, which may be considered an incorrect citation style.

General: Short and on-point. Quick and well-written introduction to the topic.]

Response1 [Thank you for your suggestion. In accordance with your guidance, we referred to the 'Instructions for Authors' and confirmed the guideline stating, 'References in the text should be placed in square brackets [ ], before the punctuation (e.g., [1], [1-3], or [1,3]).' Therefore, we have followed this guideline. Once again, thank you for thoroughly reviewing our manuscript.]

Materials and Methods:

Comment2 [Were bilateral fractures excluded?

Please describe the types of pelvic fractures included. Which classification system was used?

General: Except for the aforementioned aspects, this is a well-written section.]

Response2 [In the field of trauma, pelvic fractures are primarily categorized into three main types according to the Young-Burgess classification: anterior-posterior compression (APC), lateral compression, and vertical shear injury. However, we were not able to investigate the type of pelvic fracture, which is one of the limitations of this study. This study focused on patients with hemorrhagic fractures rather than the types of pelvic fractures and included all patients with hemorrhagic fractures during the study period. We have mentioned this point in lines 275 to 278 of the Discussion section.]

Results:

Comment3 [Line 144: What were the reasons for reimplementation?]

Response3 [Since preperitoneal pelvic packing is a temporary method, a reoperation is required 1-2 days after the initial surgery to remove surgical pads. If it is determined that bleeding is not sufficiently controlled during the reoperation, reimplementation is performed. I have added the details of this process to lines 112 to 115 of the manuscript.]

Discussion:

Comment4 [Line: 168-189: this section is redundant and wordy. PPP is primarily performed in unstable patients. Please shorten the whole paragraph. (Line 186-194)]

Response4 [We totally agree with you. We have revised the text for more conciseness.]

Comment5 [Line: 195-215: this section is more suitable for the introduction section and not the discussion section. Please rewrite and shorten it significantly and discuss the findings mentioned in the manuscript.]

Response5 [We strongly agree with you. We have condensed this section and rearranged some content considering the context of the text, as per your suggestion. Thank you for your valuable input.]

Round 2

Reviewer 1 Report

Comments and Suggestions for Authors

I thank the authors for their efforts to consider the suggestions made by the reviewers.

Author Response

Comment: I thank the authors for their efforts to consider the suggestions made by the reviewers.

Response: Thank you for your kind words and for recognizing our efforts in addressing the suggestions provided. We greatly appreciate the valuable feedback from you and the other reviewers, which has significantly improved the quality of our manuscript. Your constructive comments and insights were instrumental in refining our research. We are grateful for the time and effort you have dedicated to reviewing our work and for your positive acknowledgment of our revisions.

Reviewer 2 Report

Comments and Suggestions for Authors

The authors failed to address all of my requests. Point 1 and 2 are not improved or changed. Their explanation has nothing to with the changes that have to been made. Fracture type is an important part, as stability and further treatment is depending on it. 

Author Response

Comment:

The authors failed to address all of my requests. Point 1 and 2 are not improved or changed. Their explanation has nothing to with the changes that have to been made. Fracture type is an important part, as stability and further treatment is depending on it.

 Response: Despite your kind remarks, we apologize for any shortcomings in our revisions.

We have re-examined point 1 and made the necessary corrections, which can be found on line 35.

Regarding point 2, We completely agree with your observations. Thank you for your insightful comments. We have re-evaluated the points you raised. Following your recommendation, we re-investigated the pelvic fracture type for all patients. These fractures were classified according to the Young & Burgess classification, and this process involved more than two surgeons. We conducted a statistical analysis of the pelvic fracture types between the conventional PPP group and the modified PPP group. There was no statistically significant difference in the fracture types between the two groups. This information has been reflected in Table 1. Thank you again for reviewing our work and providing your valuable feedback. Please do not hesitate to let us know if there are any additional concerns or suggestions.